# Evaluating Cognitive-Motor Interference in Multiple Sclerosis: A Technology-Based Approach

**DOI:** 10.3390/bioengineering11030277

**Published:** 2024-03-14

**Authors:** Jessica Podda, Ludovico Pedullà, Giampaolo Brichetto, Andrea Tacchino

**Affiliations:** 1Scientific Research Area, Italian Multiple Sclerosis Foundation, 16149 Genoa, Italy; jessica.podda@aism.it (J.P.); ludovico.pedulla@aism.it (L.P.); giampaolo.brichetto@aism.it (G.B.); 2AISM Rehabilitation Service of Genoa, Italian Multiple Sclerosis Society, 16149 Genoa, Italy

**Keywords:** multiple sclerosis, cognitive impairment, single-task, dual-task, cognitive-motor interference, dual-task cost, wearable, semantic fluency

## Abstract

Background: People with multiple sclerosis (PwMS) frequently present both cognitive and motor impairments, so it is reasonable to assume they may have difficulties in executing dual-tasks (DT). The aim of the present study is to identify novel technology-based parameters to assess cognitive-motor interference (CMI) in PwMS. In particular, we focused on the definition of dual-task cost (DTC) measures using wearable and portable tools such as insoles and mobile apps. Methods: All participants underwent a verbal fluency task (cognitive single-task, ST), a motor ST of walking, and a combination of these tasks (DT). Number of words uttered in the cognitive ST and steps recorded by insoles were used to calculate the motor and cognitive DTC. Results: The number of steps strongly correlated with the walked meters for both single- (r = 0.88, *p* < 0.05) and dual- (r = 0.91, *p* < 0.05) tasks. Motor but not cognitive performances significantly worsened during DT. Over the cognitive ST and DT, the number of pronounced words progressively decreased, probably due to the activation of different cognitive processes. Cognitive efforts could be the cause of cognitive task prioritization. Conclusions: Our findings promote the use of low-cost devices to assess CMI easily in the clinical context and to detect ecologically valid DT impairments.

## 1. Introduction

Real-world walking rarely occurs in isolation as it is often accompanied by increased attentional demands based on performing simultaneous tasks. Irrespective of having any sensory or motor disorders, individuals with cognitive impairments (CIs) pose a relatively higher risk of falling compared to those without CIs. Thus, the simultaneous performance of cognitive and motor tasks can be difficult and can lead to worse performance in the cognitive or motor domain or both [1]. The dual-task (DT) paradigms are one of the most common ways to gauge this interaction and allow for so-called cognitive-motor interference (CMI) to be revealed, occurring when performance in a motor or cognitive task decreases on performing a DT as compared to a single-task (ST) [2]. CMI is usually quantified as DT cost (DTC), which is the percentage change from ST to DT performance [3]. 

People with multiple sclerosis (PwMS) frequently present both cognitive and motor impairments, and it is well documented that they face DT difficulties [4]. Although some studies have reported similar DTC between PwMS and healthy adults [5], the impact of CIs on motor performance has been proven among PwMS [6], with possible negative consequences on their daily living activities [7,8]. For example, a simple arithmetic task executed during balance maintenance led PwMS to increased postural sway and a large decrement in the variability of antero-posterior and medio-lateral sway velocity [9,10,11]. Concerning ambulation tasks, PwMS showed increased stride time and decreased speed while walking under several cognitive conditions such as talking [12,13,14], as a possible result of a divided attention deficit [7].

Although comprehensive results from a plethora of studies showed worse motor performance during DT in PwMS, reflecting muscle fatigue, poor coordination, and spasticity resulting from the variability in gray matter atrophy, they found that DT testing is not able to measure ‘real life’ issues, e.g., could not predict future falls among PwMS [9]. 

Thus, increasingly, investigators are attempting to understand the underlying mechanisms of CMI in PwMS during walking and design DT paradigms directed towards meeting demands of ‘real life’ situations [15]. Technological tools are helpful since they provide a solution to the limitations of current clinical and lab-based measures by enabling easy-to-administer continuous monitoring in a clinical context or outside of a typical laboratory [16]. Considering both cognitive and motor domains, wearable devices represent promising tools for enhancing real-time data capture to screen for, monitor, and treat motor and cognitive difficulties in MS [17,18] that may fundamentally shift clinical traditional paradigms of medicine [19]. 

The use of these digital alternatives brings potential advantages such as easiness to administer, continuous data collection, and high sensitivity to subtle changes in impairment [20]. The DT literature in MS shows that inertial measurement units (IMUs) (e.g., WALK-MATE Viewer, WALK-MATE LAB., Tokyo, Japan; Opal sensors, APDM, Portland, OR, USA; IMUs/MTw, Xsens Technologies BV, Enschede, the Netherlands; G-Sensor^®^, BTS Bioengineering S.p.A., Milan, Italy) [4,7] and fNIRS (e.g., fNIR Imager 1000, fNIR Devices LLC, Potomac, MD, USA) are the mainly used wearable devices. However, if fNIRS could add relevant insights on DT performances through the analysis of hemodynamic activity, the assessment with IMUs is limited to spatial–temporal and kinematic parameters of gait, for example neglecting other interesting information such as forces. Furthermore, in both cases, they are not low-cost solutions and, even if they represent the main options for the DT assessment in a laboratory context, their use in a real-world context is not feasible. Some dedicated apps had been used for the assessment of cognitive performances during DT in MS (e.g., CMI-APP) [19]; however, they are not available or downloadable from the app stores.

The aim of the present study was to identify novel technology-based parameters to assess CMI in PwMS. In particular, we preliminarily tested the feasibility of a paradigm to gauge new DTC outcomes (e.g., peak of forces) using wearable and portable tools such as insoles and mobile apps, ensuring an easy-to-use and ecological format. 

## 2. Materials and Methods

### 2.1. Participants

People diagnosed with clinically defined multiple sclerosis (MS) according to the McDonald criteria [21] were recruited among those followed as outpatients at the Italian Multiple Sclerosis Society (AISM) Rehabilitation Service of Genoa. Inclusion criteria were an Expanded Disease Status Scale (EDSS) score ranging from 0 to 6.0 [22], all disease courses (i.e., relapsing–remitting, RR; primary progressive, PP; and secondary progressive, SP), a disease-stable phase without relapses in the last 3 months, and adequate visual, hearing, and motor capabilities to work on a smartphone or a tablet. Exclusion criteria were a Montreal Cognitive Assessment score < 18, neurological and major psychiatric illness, past serious head trauma, and alcohol or drug abuse, pregnancy, or being unable to comply with the requirements of the protocol. All subjects provided written, informed consent. The project was approved by the local Ethics Committee, and all patients signed an informed consent form prior to their inclusion in the study.

### 2.2. Experimental Protocol

All participants underwent a cognitive ST, a motor ST, and a combination of these tasks (DT). The order in which the blocks were performed was randomized.

As the cognitive ST (ST_cogn_), we used the semantic variant of the Word List Generation (WLG) [23,24]; it consists of naming in 90 s as many words as possible belonging to a certain category (randomly chosen among animals, fruits, and vegetables). The score is the number of correct words (WORDS) pronounced for the specific class.

As the motor ST (ST_mot_), we asked the participants to walk at their preferred velocity over a 10 m long corridor. The score is the distance walked in 90 s (DIST) in meters, as recorded by the therapist.

The DT consisted of simultaneously performing both single tasks and lasted 90 s. Participants were not given any instructions regarding the prioritization of the walking or semantic task, but they were asked to perform both tasks as best as they could. 

Subjects paused for 1 min between trials to allow time for the assessor to set up the next trial and the participant to rest. Prior to initiation, participants received instructions on and conducted familiarization with the ST and DT. 

### 2.3. Devices

The WLG was performed through the “Generate words” test of DIGICOG-MS^®^ (SIAE Registration ID: D000018162, v 1.0, 27 December 2022), a mHealth app developed by the Italian Multiple Sclerosis Foundation to self-assess and self-monitor the presence of CIs in PwMS. The app, supported on Android and iOS and developed for data collection and data presentation to the user, implements a cloud service for data storage, analysis algorithms, and a clinician dashboard for user management and data extraction. Machine learning algorithms for the words’ detection will be implemented in future releases of the app, but for the aims of this study, a neuropsychologist had processed recordings of pronounced words to score participants’ performance. The mHealth App Usability Questionnaire (MAUQ) was administered to test the usability of DIGICOG-MS^®^ (18 items; item score range: 1–7; total score range: 18–126; usability cut-off: 72; subscales: ease of use, interface and satisfaction, and usefulness) [25].

Besides the meters walked in 90 s, motor performance was also assessed with the Loadsol insoles (https://www.novel.de/products/loadsol/ (accessed on 13 March 2024), version with two sensors at the level of the forefoot and heel). Powered by coin batteries, they transmit data via Bluetooth to a smartphone. The sensor accurately measures the normal plantar force detected inside the shoe during all static and dynamic activities using a thin insole, which does not disturb the proprioception of the foot. The force values are stored on the smart device and to the cloud, displayed on the smartphone in real time, and additionally transferred to a computer for a more detailed analysis.

Already used to study walking and running in real-world settings [26], these insoles are able to observe plantar forces and can facilitate more comprehensive assessments and novel insights into the kinetic characteristics of PwMS during dual-task conditions in both clinical and ecological contexts.

For both feet, the following parameters were automatically calculated through the Loadpad analysis software (v 28.3.8.6):Number of steps (STEPS), i.e., the sum of steps executed with both feet.Average contact time (CT, ms), i.e., the contact time averaged over all the steps.Average peak force (PF, N), i.e., the peak force averaged over all the steps.Force time integral (FTI, Ns), i.e., force–time integral over the entire interval of the trial.

The CT, PF, and FTI were also automatically calculated for both feet and both sensors (i.e., forefoot and heel).

Moreover, the Factor of Imbalance (FOIB) was calculated based on the FTI of both insoles:FOIB = (FTI_left_ − FTI_right_)/(FTI_left_ + FTI_right_); it measures which foot is loaded more over the interval of analysis. Thus, negative or positive values mean that the right or the left foot was loaded more, respectively; here, we used the absolute value of the FOIB in order to describe imbalance more generally.

Finally, in order to evaluate if differences were present during the ST and DT, we evaluated changes in WORDS and STEPS every 30 s (i.e., WORDS_30_, WORDS_60_, WORDS_90_; STEPS_30_, STEPS_60_, STEPS_90_).

### 2.4. Calculating the Dual-Task Cost (%)

The DTC was calculated according to the formula of Baddeley et al. (1997) [27] for each subject as follows:Motor DTC (DTC_mot_) = (ST_mot_ − DT_mot_)/ST_mot_×100;Cognitive DTC (DTC_cogn_) = (ST_cogn_ − DT_cogn_)/ST_cogn_×100.

The DTC_mot_ was calculated for both DIST and STEPS; the DTC_cogn_ was obtained from the uttered WORDS. Positive DTC values indicate lower DT ability, whereas a negative value indicates higher DT ability. The magnitude of DTC_mot_ and DTC_cogn_ was calculated for all participants.

### 2.5. Statistical Analysis

The Shapiro–Wilk test was used to assess normality in the continuous variables; all variables resulted in being normally distributed. DIST, STEPS, WORDS, FOIB, and all the other parameters obtained from the insoles were subjected to a repeated-measure analysis of variance by taking into consideration the condition (ST, DT) and the foot (Right, Left) as within-subject factors. Disease duration (DUR) and sex (SEX) were included as covariates in the analysis. Post hoc analysis was performed with the Bonferroni test. A sub-analysis comparing the participants for different DUR (based on median value) and SEX was also similarly performed. 

Pearson’s test was used to correlate DTC_mot_ calculated on DIST and STEPS. Correlation was considered as low for r < 0.30; moderate for r: 0.30–0.59; high for r: 0.60–0.79; and very high for r ≥ 0.80.

All statistical tests were used with a two-tailed analysis and 0.05 as a level of significance. 

The statistical software STATISTICA (v 7.1, StatSoft GmbH, Tulsa, OK 74104, USA) was used for all analyses.

## 3. Results

### 3.1. Participants

In total, 23 PwMS participated in the study (15 females and 8 males; mean age 29.7 ± 8.4 years, range: 19–61 years). The mean EDSS was 2.2 ± 1.6 (range: 0–5.5); the average disease duration was 4.2 ± 4.9 years (range: 1–20 years, median: 3.0); 20 were RR, 2 SP, and 1 PP; none had comorbidities. All the participants concluded high school, seven were a graduate, and six attended university. 

### 3.2. Cognitive Performance

No significant differences were found between ST_cogn_ (23.91 ± 8.37) and DT (21.04 ± 6.81) in WORDS (*p* = 0.11). Similarly, no significant differences were found between conditions when participants were grouped for DUR (group: *p* = 0.95; condition: *p* = 0.21; interaction: *p* = 0.40) and SEX (group: *p* = 0.47; condition: *p* = 0.22; interaction: *p* = 0.83). A positive DTC_cogn_ (6.40 ± 34.72) seems to suggest a worsening trend in performances in DT.

The analysis of words uttered in the different phases of the trial showed significant differences (always *p* < 0.001) between WORDS_30_ and both WORDS_60_ and WORDS_90_ for both ST and DT (ST: WORDS_30_ = 13.09 ± 3.68, WORDS_60_ = 7.17 ± 3.33, and WORDS_90_ = 3.65 ± 3.02; DT: WORDS_30_ = 11.83 ± 3.23, WORDS_60_ = 5.39 ± 2.35, and WORDS_90_ = 3.83 ± 2.96). Significant differences were present between WORDS_60_ and WORDS_90_ (*p* < 0.01) only in ST (Figure 1A). No significant differences were present in WORDS_30_, WORDS_60_, and WORDS_90_ between ST and DT; significant differences were present between WORDS_30_ in ST and WORDS_60_, and WORDS_90_ in DT, and between WORDS_60_ in ST and WORDS_90_ in DT.

The total score from MAUQ was 112.96 ± 10.72, suggesting that DIGICOG-MS^®^ was usable and well appreciated by PwMS (subscales: ease of use: 32.60 ± 3.31; interface and satisfaction: 45.26 ± 4.03; usefulness: 35.08 ± 5.03).

### 3.3. Motor Performance

During the ST_mot,_ participants walked on average for 143.15 ± 37.73 m by performing 172.09 ± 25.57 steps (r = 0.88, *p* < 0.05); during the DT, the performances worsened, and they walked on average for 128.80 ± 37.98 m by performing 160.74 ± 27.23 steps (r = 0.91, *p* < 0.05). Significant differences between ST_mot_ and DT were present for both DIST (*p* < 0.001) and STEPS (*p* < 0.001) (Table 1). When participants were grouped for DUR and SEX for DIST, significant differences were found between conditions (in both cases, *p* < 0.001) but not between groups (*p* = 0.82 and *p* = 0.58, respectively) and in the interaction (*p* = 0.30 and *p* = 0.17, respectively). Similarly, by analyzing STEPS after grouping for DUR and SEX, significant differences were found between conditions (in both cases, *p* < 0.001) but not between groups (*p* = 0.45 and *p* = 0.08, respectively) and in the interaction (*p* = 0.78 and *p* = 0.07, respectively). As expected, a steps reduction in DT was reflected in the longer contact time (Table 1); indeed, CT in DT was significantly longer than in ST_mot_ (*p* = 0.029). Significant differences were found between feet in ST (*p* < 0.01) but not in DT (*p* = 0.14). Similar results were found by considering the CT at the level of the forefoot. No significant differences were found at the heel level. When participants were grouped for DUR and SEX, significant differences were found between conditions (in both cases, *p* < 0.01) but not between groups (*p* = 0.43 and *p* = 0.98, respectively) and in the interactions. Similar results were found when the analysis was performed at the level of the forefoot, whereas no differences were found at the level of the heel.

The worsening in the performances was also represented by a positive DTC_mot_, as based on both DIST (10.82 ± 6.57) and STEPS (6.83 ± 5.02) (Table 1). Importantly, also the correlation between these two DTC_mot_ was high and significant (r = 0.79, *p* < 0.05), suggesting a reliable use of insoles to automatically record motor performances also in DT conditions.

The analysis of steps performed in the different phases of the trial did not show significant differences between STEPS_30_, STEPS_60,_ and STEPS_90_ in both ST and DT (ST: STEPS_30_ = 56.70 ± 8.34, STEPS_60_ = 56.09 ± 8.85, and STEPS_90_ = 55.91 ± 8.69; DT: STEPS_30_ = 53.09 ± 9.26, STEPS_60_ = 52.57 ± 9.06, and STEPS_90_ = 51.83 ± 9.37) (Figure 1B). Significant differences were present between ST and DT among all the different phases (always *p* < 0.001).

### 3.4. Load Forces

The analysis of plantar forces showed significant differences between ST_mot_ and DT in PF (*p* < 0.01) (Table 1); no differences were found between left and right feet for both conditions. Similar results (*p* < 0.01) were found by considering the PF at the level of the heel. No differences were found for foot and condition at the level of the forefoot. When participants were grouped for DUR, significant differences were found between conditions (*p* < 0.01) but not for any other main effects (i.e., group and feet) and interaction; similar results were found at the level of the heel; and no significant effects were found at the level of the forefoot. By grouping for SEX, significant differences were found between conditions (i.e., ST_mot_ and DT) (*p* < 0.001) and between females and males (727.27 ± 131.78 N and 856.51 ± 103.96 N, respectively; *p* < 0.05); significant differences at the level of the heel were only found between conditions (*p* < 0.001); and no significant differences were found at the level of the forefoot. No significant differences were present between ST_mot_ and DT in FTI and in FTI at the level of the heel and forefoot (Table 1). When participants were grouped for DUR, no significant differences were found for any comparison (i.e., group, condition, and feet) and interaction. By grouping for SEX, only a significant difference (*p* < 0.05) was found between females (28,176.50 ± 4487.79 Ns) and males (34,651.88 ± 6144.99 Ns). Similar results were found when the analyses were performed at the level of the forefoot and heel.

No significant differences were present between ST_mot_ and DT_mot_ in FOIB; this result was also confirmed when the participants were grouped for DUR and SEX.

## 4. Discussion

CIs are recognized as one of the most disturbing disorders in MS [28]. While walking, CMI could alter both the walking pattern (e.g., gait velocity) and the cognitive task performance (e.g., verbal fluency).

Here, we digitally administered a WLG test through DIGICOG-MS^®^; this app, already user-tested in MS, was considered highly usable by the participants. Furthermore, we used the Loadsol insoles to evaluate motor performances.

The number of steps correlated very highly with the walked meters in both ST and DT and, as expected, high significant correlation was found between DTC_mot_ obtained using steps and meters. Thus, the number of steps could be proposed as a new automatic way to measure CMI in the motor domain. Moreover, this new paradigm testing DT with wearables and portable tools may allow for DTC assessment in ecological conditions and provide an objective measure of the impact of DT difficulties, as reported by PwMS [29].

Motor performances significantly worsened in DT condition, as reported elsewhere [4], whereas no significant changes were found in the generation of words between task conditions. This result would reflect a prioritization towards the semantic fluency task [4].

However, it is interesting to notice the significant decrement of generated words across time in both ST and DT. This is in line with previous evidence proving that, throughout a fluency task, word production rate decreases regardless of the presence of Cis, and less common words are generated at the end of the task [30]. Following the Smith and Claxton’s lexical organization model [31], individuals initially access the “topicon”, their long-term store, containing commonly used and easily accessible words; when this stock is utilized, individuals try to retrieve words from the larger lexicon, which requires more strenuous effort [32]. Thus, different cognitive processes would moderate the fluency task over time: information processing speed at the early semi-automatic words retrieval stage followed, at later stages, on from a more effortful retrieval from the larger semantic memory store. 

Due to the required effort and the change in the active cognitive processes, during DT, participants would prioritize the cognitive task; as a consequence, although no changes were found over time in meters and steps in both ST and DT, a decline during DT occurred in motor performances. 

The analysis of forces observed by the insoles showed a significantly lower peak of forces, with specific reference to the heel, during DT; this reflects a slower walk. It could be of interest for therapists when load management is the main goal of a rehabilitative intervention (e.g., ambulation). 

Our study has some limitations (e.g., small sample size and lack of comparison to healthy individuals) given the explorative nature of this study. Moreover, as proof-of-concept, we investigated only one cognitive test; therefore, as different types of cognitive tasks may result in different patterns of CMI, we suggest to couple, in future studies, tests involving other cognitive domains.

## 5. Conclusions

Our findings promote the use of low-cost devices to assess CMI easily in the clinical context and to detect ecologically valid DT impairments. An automatic recording of cognitive-motor performances during DT could provide more ready-to-use data for the definition of rehabilitation strategies to adopt. However, more research is required.

## Figures and Tables

**Figure 1 bioengineering-11-00277-f001:**
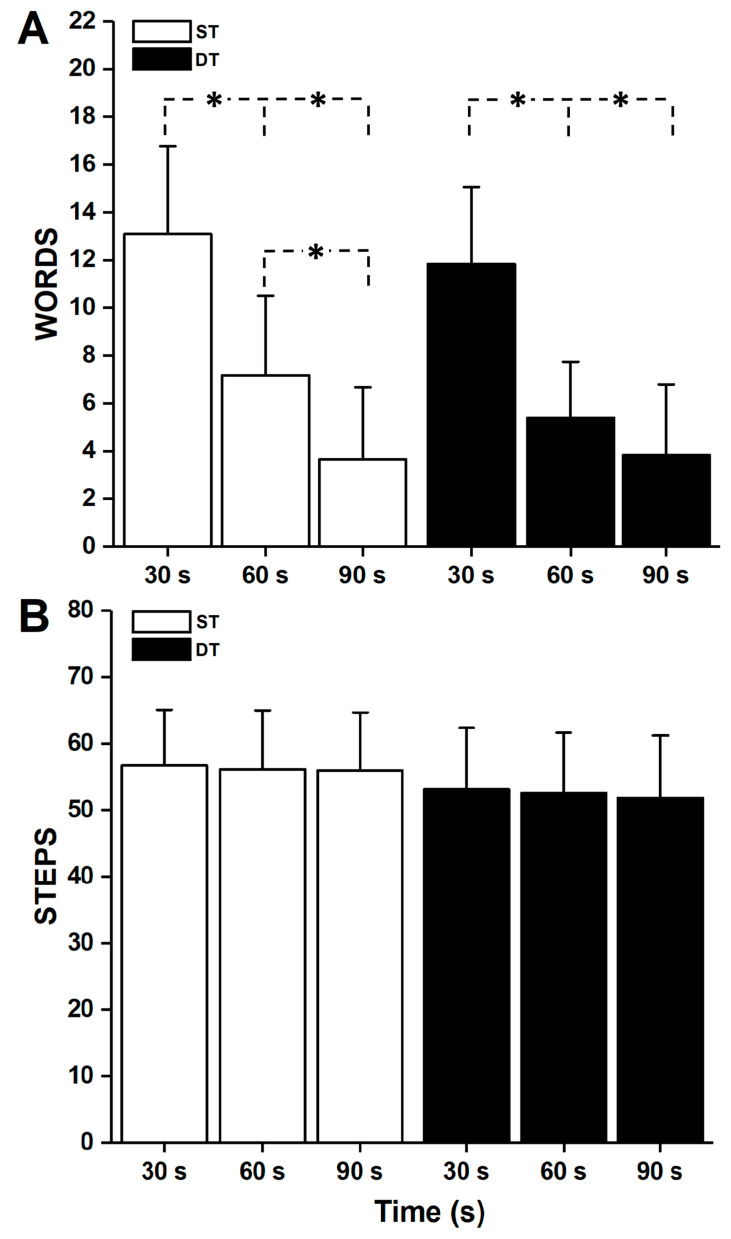
Changes in WORDS (**A**) and STEPS (**B**) every 30 s for both single- and dual-tasks. Significance, indicated with *, is only shown for statistical differences between the single-task and dual-task.

**Table 1 bioengineering-11-00277-t001:** Motor and cognitive parameters obtained from devices used during single- and dual-tasks.

	Single-Task (ST)		Dual-Task (DT)		*p* Value
	Right	Left	Right	Left	
Contact time (ms)—CT					
– foot	682.91 ± 140.54 ^a,b,c^	702.50 ± 182.60 ^a,d,e^	736.50 ± 199.43 ^b,d^	747.18 ± 215.53 ^c,e^	0.029
– heel	462.27 ± 150.58 ^c^	486.91 ± 181.27	495.55 ± 203.72	513.91 ± 212.94 ^c^	0.091
– forefoot	593.18 ± 133.70 ^a,b,c^	609.77 ± 181.42 ^a,d,e^	643.45 ± 188.32 ^b,d^	652.27 ± 212.73 ^c,e^	0.016
Peak force (N)—PF					
– foot	786.61 ± 135.39 ^b,c^	789.72 ± 147.67 ^d,e^	755.92 ± 127.49 ^b,d^	757.85 ± 141.43 ^c,e^	<0.01
– heel	606.81 ± 123.42 ^a,b,c^	629.69 ± 125.30 ^a,d,e^	569.56 ± 117.89 ^b,d^	585.25 ± 112.99 ^e^	<0.01
– forefoot	722.36 ± 147.58	728.82 ± 147.40	707.39 ± 144.74	705.08 ± 139.20	0.27
Force time integral (Ns)—FTI					
– foot	29,948.73 ± 5136.25	31,109.76 ± 7207.41	29,983.56 ± 5084.11	30,673.20 ± 6467.27	0.27
– heel	12,051.92 ± 1900.55	12,769.76 ± 3103.62	11,837.63 ± 2466.11	12,295.68 ± 3390.92	0.57
– forefoot	17,855.42 ± 4089.36	18,380.50 ± 5113.99	18,187.32 ± 4050.62	18,337.01 ± 4777.71	0.95
	**Single-Task (ST)**	**Dual-Task (DT)**	***p* value**		**Dual-Task Cost (DTC)**
Walked distance (m)—DIST	143.15 ± 37.73	128.80 ± 37.98	<0.001		10.82 ± 6.57
Number of steps—STEPS	172.09 ± 25.57	160.74 ± 27.23	<0.001		6.83 ± 5.02
Generated words—WORDS	23.91 ± 8.37	21.04 ± 6.81	0.11		6.40 ± 34.72
Factor of Imbalance (%)—FOIB	3.5 ± 6.3	3.0 ± 3.3	0.70		

^a, b, c, d, e^ indicate statistically significant differences at the post hoc analysis (^a^ between ST right and ST left; ^b^ between ST right and DT right; ^c^ between ST right and DT left; ^d^ between ST left and DT right; ^e^ between ST left and DT left.

## Data Availability

The data supporting the findings of this study are available from the corresponding authors upon reasonable request.

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
