# Peer review of "Evaluating Cognitive-Motor Interference in Multiple Sclerosis: A Technology-Based Approach"

_bioengineering, 2024, doi:10.3390/bioengineering11030277_

Round 1

Reviewer 1 Report

Comments and Suggestions for Authors

In this work, the authors evaluate the validity of wearable sensors and mobile apps to assess cognitive-motor interference in patients with multiple sclerosis (MS). Using insoles to collect information on walking parameters and mobile apps to propose a verbal fluency task, they show that motor but not cognitive performances are altered by dual task, and that the number of steps strongly correlate with the walked meters. They conclude that low-cost devices can be used to assess cognitive-motor interference in MS patients.

It is difficult to evaluate the novelty of this work since cognitive-motor interference (CMI) in MS patients is already well documented in the literature, and wearable device / apps are also used in these patients.

 The authors fail to highlight the importance, the added-value of their work, because the introduction does not make clear:                       -  If they want to demonstrate CMI in MS patients

-       It they want to validate the use of insoles and apps to reveal CMI

-       why measuring CMI at home is crucial for patients and how it can provide interesting information on underlying mechanisms

The paper would benefit rewriting the introduction to better explain what is already known, already done, already available… what is the objective of the study and novelty of the work…

A lot of references are missing to help reaching this issue, I suggest few here (not exhaustive list):

Wallin et al., Mult Scler Relat Disord 2022

Cognitive-motor interference in people with mild to moderate multiple sclerosis, in comparison with healthy controls DOI: 10.1016/j.msard.2022.104181

Veldkamp et al., J Neurol 2023

Profiling cognitive-motor interference in a large sample of persons with progressive multiple sclerosis and impaired processing speed: results from the CogEx study DOI: 10.1007/s00415-023-11636-y

Block et al. JNeurol 2017

Continuous daily assessment of multiple sclerosis disability using remote step count monitoring DOI: 10.1007/s00415-016-8334-6

Schleimer et al., J Med Internet Res 2020

A Precision Medicine Tool for Patients With Multiple Sclerosis (the Open MS BioScreen): Human-Centered Design and Development DOI: 10.2196/15605

Montalban et al., Mult Scer 2022

A smartphone sensor-based digital outcome assessment of multiple sclerosis DOI: 10.1177/13524585211028561

Reviewer 2 Report

Comments and Suggestions for Authors

I thank the editor for the opportunity to review this article. It is very interesting, however I think the authors must resolve some important points before it is accepted:

1.      There are several sentences that do not have bibliographic references. Please add them. For ex. page 1, line 38 to 40 "Indeed, some reviews showed worse motor performance during DT indicating motor interference as primary effect." Review the entire text and do not leave statements based on literature without citation.

2.      I suggest that authors use a checklist to write the manuscript. For this type of study, STROBE seems to be the most appropriate.

3.      In the results, use the number of decimals that seems reasonable. For example, the age is 29.70 +/- 8.44. Here just put 29.7 +/- 8.4

4.      I miss a table 1 with the description of the characteristics of the population. Although the authors describe the population in the subtitle participants, I would like to know other characteristics such as BMI, presence of comorbidities, etc.

5.      The range of duration of the disease is very wide (from 1 to 20 years). Also the severity of the disease can vary greatly between a subject of 1 and another of 20 years of progression. For that reason, it would be interesting to know if the authors can do a subanalysis according to time of progression (perhaps using the median to divide it) or perhaps there are differences associated with sex?
